# Increased Plasma Soluble PD-1 Concentration Correlates with Disease Progression in Patients with Cancer Treated with Anti-PD-1 Antibodies

**DOI:** 10.3390/biomedicines9121929

**Published:** 2021-12-16

**Authors:** Ryotaro Ohkuma, Katsuaki Ieguchi, Makoto Watanabe, Daisuke Takayanagi, Tsubasa Goshima, Rie Onoue, Kazuyuki Hamada, Yutaro Kubota, Atsushi Horiike, Tomoyuki Ishiguro, Yuya Hirasawa, Hirotsugu Ariizumi, Junji Tsurutani, Kiyoshi Yoshimura, Mayumi Tsuji, Yuji Kiuchi, Shinichi Kobayashi, Takuya Tsunoda, Satoshi Wada

**Affiliations:** 1Department of Clinical Diagnostic Oncology, Clinical Research Institute for Clinical Pharmacology & Therapeutics, Showa University, 6-11-11, Kitakarasuyama, Setagaya-ku, Tokyo 157-8577, Japan; rohkuma@med.showa-u.ac.jp (R.O.); kieguchi@med.showa-u.ac.jp (K.I.); mwata@cnt.showa-u.ac.jp (M.W.); dtakayan@med.showa-u.ac.jp (D.T.); tsubasa-g@med.showa-u.ac.jp (T.G.); onoue.r@cnt.showa-u.ac.jp (R.O.); 2Department of Medicine, Division of Medical Oncology, School of Medicine, Showa University, 1-5-8, Hatanodai, Shinagawa-ku, Tokyo 142-8555, Japan; hamadakaz@med.showa-u.ac.jp (K.H.); yutaro@med.showa-u.ac.jp (Y.K.); horiike@med.showa-u.ac.jp (A.H.); tomo-1496@med.showa-u.ac.jp (T.I.); h.yuya0204@med.showa-u.ac.jp (Y.H.); ariizumi@med.showa-u.ac.jp (H.A.); tsurutaj@med.showa-u.ac.jp (J.T.); kyoshim1@med.showa-u.ac.jp (K.Y.); ttsunoda@med.showa-u.ac.jp (T.T.); 3Clinical Research Institute for Clinical Pharmacology & Therapeutics, Showa University, 6-11-11, Kitakarasuyama, Setagaya-ku, Tokyo 157-8577, Japan; s2koba@med.showa-u.ac.jp; 4Advanced Cancer Translational Research Institute, Showa University, 1-5-8, Hatanodai, Shinagawa-ku, Tokyo 142-8555, Japan; 5Department of Clinical Immuno Oncology, Clinical Research Institute for Clinical Pharmacology & Therapeutics, Showa University, 6-11-11, Kitakarasuyama, Setagaya-ku, Tokyo 157-8577, Japan; 6Department of Pharmacology, Division of Medical Pharmacology, School of Medicine, Showa University, 1-5-8, Hatanodai, Shinagawa-ku, Tokyo 142-8555, Japan; tsujim@med.showa-u.ac.jp (M.T.); ykiuchi@med.showa-u.ac.jp (Y.K.); 7Pharmacological Research Center, Showa University, 1-5-8, Hatanodai, Shinagawa-ku, Tokyo 142-8555, Japan

**Keywords:** biomarkers, immune checkpoint inhibitors, anti-programmed death-1 (PD-1)

## Abstract

Immune checkpoint inhibitors (ICIs) confer remarkable therapeutic benefits to patients with various cancers. However, many patients are non-responders or develop resistance following an initial response to ICIs. There are no reliable biomarkers to predict the therapeutic effect of ICIs. Therefore, this study investigated the clinical implications of plasma levels of soluble anti-programmed death-1 (sPD-1) in patients with cancer treated with ICIs. In total, 22 patients (13 with non-small-cell lung carcinoma, 8 with gastric cancer, and 1 with bladder cancer) were evaluated for sPD-1 concentration using enzyme-linked immunosorbent assays for diagnostic and anti-PD-1 antibody analyses. sPD-1 levels were low before the administration of anti-PD-1 antibodies. After two and four cycles of anti-PD-1 antibody therapy, sPD-1 levels significantly increased compared with pretreatment levels (*p* = 0.0348 vs. 0.0232). We observed an increased rate of change in plasma sPD-1 concentrations after two and four cycles of anti-PD-1 antibody therapy that significantly correlated with tumor size progression (*p* = 0.024). sPD-1 may be involved in resistance to anti-PD-1 antibody therapy, suggesting that changes in sPD-1 levels can identify primary ICI non-responders early in treatment. Detailed analysis of each cancer type revealed the potential of sPD-1 as a predictive biomarker of response to ICI treatment in patients with cancer.

## 1. Introduction

Programmed death-1 (PD-1) and its ligand, PD-L1, are central components of immune checkpoints, which are signaling pathways that regulate the immune system, preventing the immune system from attacking normal cells and maintaining immune homeostasis and self-tolerance [1,2]. PD-1 and PD-L1 are activated by ligand–receptor binding, blocking their signaling pathways; therefore, negative costimulatory molecules are abnormally expressed on tumor cells by immune checkpoints, allowing tumor cells to escape the inherent antitumor immune response [2,3]. Therefore, the tumor microenvironment is again endowed with antitumor immunity. PD-L1 is also involved in immune escape through its interaction with PD-1. Therefore, targeting PD-1/PD-L1 signaling using immune checkpoint inhibitors (ICIs) such as anti-PD-1 antibodies (nivolumab and pembrolizumab) has provided remarkable therapeutic benefits in the treatment of various cancers [4]. Evidence has shown that the PD-1/PD-L1 pathway is intimately involved in resistance to antitumor immunity in several types of cancer [5]. PD-1/PD-L1 blockade therapy is currently considered one of the most significant breakthroughs in cancer immunotherapy. Over the past few years, ICIs, including both anti-PD-1 and anti-PD-L1 antibodies, have shown favorable efficacy in both advanced non-small-cell lung cancer (NSCLC) and extensive-stage small-cell lung cancer (SCLC) [6,7]. In addition, ICI treatment could enhance survival benefits for patients with advanced gastric cancer. Since then, nivolumab has been used in unresectable and advanced gastric cancer [8]. Similarly, the efficacy of single-agent anti-PD-1 antibody has been confirmed in bladder cancer [9]. Therefore, to investigate the relationship between ICI treatment and sPD-1, we focused on anti-PD-1 antibody drugs (nivolumab and pembrolizumab), which are widely used as monotherapy.

In addition to surgery, radiotherapy, and chemotherapy, ICIs have become an essential remedy against malignant tumors. Unfortunately, there are specific patient populations for whom the effects of immunotherapy are limited. Many patients fail to respond or develop resistance following an initial response to ICIs. There are some cases in which ICIs have little or no effect. Predictive biomarkers to identify potential ICI responders are currently in discussion. In clinical practice, PD-L1 expression on tumor cells has been used as a candidate biomarker to predict therapeutic efficacy in several types of cancer. For example, several lines of evidence suggest a correlation between PD-L1 tumor expression and ICI response in various malignancies, including lung cancer, melanoma, refractory Hodgkin’s lymphoma, and other types of solid tumors [10,11,12].

Most studies on immune checkpoint proteins, such as PD-L1, have focused on the membrane-bound form. Several studies have confirmed that the expression of soluble forms of PD-1 (sPD-1) and PD-L1 (sPD-L1) can be detected in peripheral blood (plasma and serum). However, there have been fewer studies on soluble forms than on membrane-bound forms. Some studies have indicated that soluble forms of immune checkpoint proteins are detectable in normal plasma and their levels abnormally increase in the body fluids of patients with cancer. Studies also showed that soluble forms of immune checkpoint proteins might be clinically important [1]. Despite their origin and functional roles, sPD-1 and sPD-L1 have not been sufficiently studied [13].

High sPD-L1 expression is reportedly associated with poor prognosis in multiple types of malignant tumors [14], suggesting that sPD-L1 is a predictive biomarker for poor chemotherapy response. In other words, patients with lower levels of sPD-L1 may be suitable for ICI therapy. In a previous study, we measured the levels of sPD-L1 in plasma samples collected from 21 patients with NSCLC, gastric cancer, and bladder cancer who were administered anti-PD-1 antibodies. The rate of change in sPD-L1 concentration from diagnosis to post-ICI therapy was analyzed. We demonstrated that increased plasma sPD-L1 concentration was significantly correlated with tumor progression in patients administered four cycles of anti-PD-1 antibody therapy [15]. Moreover, we focused on the clinical significance of pretreatment PD-L1 expression levels in peripheral blood mononuclear cell (PBMC) subsets, such as CD3^+^, CD4^+^, CD8^+^, and CD14^+^, in patients treated with anti-PD-1 antibodies. The results demonstrated an increased proportion of the PD-L1^+^ CD14^+^ monocyte subset that significantly correlated with shorter overall survival (OS) [16]. Knowledge from these studies indicated that sPD-L1 might be a useful predictive and prognostic biomarker to identify primary responders to anti-PD-1 antibody therapy.

To date, studies of the soluble forms of PD-1 and PD-L1 in peripheral blood have primarily focused on sPD-L1 concentrations in various types of cancer, and their association with clinicopathological characteristics and prognosis [17]. Fewer studies have measured sPD-1 concentrations in patients with cancer to investigate the predictive and prognostic roles of sPD-1 [13], and there is no definitive opinion. However, sPD-1 and sPD-L1 play essential roles in the development and progression of cancer. Moreover, sPD-1 may be a novel candidate biomarker for predicting the effects of ICIs and poor prognosis.

Herein, we measured the levels of sPD-1 in plasma collected from patients with NSCLC, gastric cancer, and bladder cancer who underwent anti-PD-1 antibody therapy, and analyzed the change in sPD-1 expression from diagnosis to posttreatment to investigate the clinical implications of sPD-1 level measurement in patients with cancer.

## 2. Materials and Methods

### 2.1. Patient Selection and Ethics Statement

We retrospectively analyzed data from 22 patients (12 with first-line or previously treated NSCLC, 9 with gastric cancer, and 1 with bladder cancer) who received anti-PD-1 antibody therapy (nivolumab (240 mg) intravenously every 2 weeks or pembrolizumab (200 mg) intravenously every 3 weeks) at Showa University Hospital between January 2017 and April 2019. For lung cancer cases, PD-L1 expression was evaluated in terms of the tumor proportion score (TPS). Pembrolizumab was selected for patients with PD-L1 expression (TPS) of 1% or higher, whereas nivolumab was available regardless of PD-L1 expression. For gastric cancer and bladder cancer, anti-PD-1 antibody drugs are indicated regardless of PD-L1 expression; therefore, PD-L1 expression was not measured. Patients were diagnosed with stage IV disease according to the Union for International Cancer Control Tumor–Node–Metastasis classification (seventh edition), or recurrent disease after surgical resection or chemoradiotherapy. Plasma levels of sPD-1 were evaluated before treatment and after two and four cycles of anti-PD-1 antibody therapy (Figure 1A). The relative change in sPD-1 concentration was determined by calculating and comparing the sPD-1 concentrations before treatment and after two and four cycles of anti-PD-1 antibody therapy. Progression-free survival (PFS) was defined as the time from anti-PD-1 antibody therapy to disease progression or death from any cause. OS referred to the time from diagnosis to the date of last follow-up or death from any cause. According to the Response Evaluation Criteria in Solid Tumors (version 1.1) [18], the efficacy of anti-PD-1 antibody therapy was assessed. Target lesions were assessed by computed tomography. Change in tumor size was calculated as the percentage change in tumor size from baseline to that after four cycles of anti-PD-1 antibody therapy. The control group comprised samples collected from six healthy volunteers. The study protocol was approved by the Ethics Committee of Showa University School of Medicine, Tokyo, Japan (approval numbers: 2165 and 2253). The research was conducted in accordance with the Declaration of Helsinki. All patients provided written informed consent prior to participation in this study.

### 2.2. sPD-1 Detection

We collected peripheral blood samples before and after ICI therapy. The plasma levels of sPD-1 were measured by enzyme-linked immunosorbent assay (ELISA) (Human PD-1 DuoSet^®^ ELISA Development System (DY1086) and DuoSet^®^ Ancillary Reagent Kit 2 (DY008); R&D Systems Inc., Minneapolis, MN, USA), according to the manufacturer’s instructions. Standards and samples were prepared as follows. Recombinant human PD-1 was diluted with 1% bovine serum albumin (BSA) in phosphate-buffered saline (PBS) for the standard curve. Plasma was centrifuged, and the supernatant was diluted 1:4 with 1% BSA. A flat-bottom 96-well microplate was coated with a 1.0 µg/mL mouse anti-human PD-1 capture antibody in PBS. The plate was sealed with an adhesive strip, followed by overnight incubation. Thereafter, the plate was washed and blocked with 1% BSA in PBS for 1 h. After washing, standards or samples were added to each well, and the plate was sealed. Two hours after incubation, the plate was washed. Thereafter, 200 ng/mL biotinylated goat anti-human PD-1 detection antibody in PBS containing 1% BSA (R&D Systems Inc., Minneapolis, MN, USA) was placed in each well. The plate was sealed and incubated for 2 h. After washing, streptavidin-horseradish peroxidase (1:200) was added to each well for colorimetric detection, the plate was sealed and then incubated for 20 min in the dark. After the plate was washed, a substrate solution consisting of a 1:1 mixture of H_2_O_2_ and tetramethylbenzidine was placed in each well and incubated for 20 min in a dark room. A termination solution was then added to the wells. The absorbance of each well was analyzed using a microplate reader (wavelength: 450 and 570 nm) (Synergy HTX; BioTek Instruments Inc., Winooski, VT, USA). The reading at 570 nm was subtracted from the reading at 450 nm to correct for optical imperfections in the plate. sPD-1 concentrations were determined using a calibration curve. The minimum detectable concentration of sPD-1 was 7.47 pg/mL.

### 2.3. Immunohistochemical (IHC) Analysis of PD-L1 Expression on Tumor Cells

We collected tumor biopsy tissues before treatment and prepared formalin-fixed paraffin-embedded tissue samples. Companion diagnostic PD-L1 IHC assays were performed: PD-L1 IHC 28-8 PharmDX and PD-L1 IHC 22C3 PharmDX assays were used before nivolumab or pembrolizumab therapy (Dako, Glostrup, Denmark), according to the manufacturer’s instructions. Two investigators were blinded to the clinical outcome, and independently evaluated specimens were stained in serial sections. PD-L1 expression was quantitatively evaluated as TPS.

### 2.4. Statistical Analyses

Statistical analyses were performed using Microsoft Excel Office 2019 (Microsoft Corp., Redmond, WA, USA). The validity of the results was confirmed using JMP version 14.0 (SAS Institute, Cary, NC, USA). The data of sPD-1 concentration are presented as median and interquartile range. The non-parametric Wilcoxon test was performed for the comparison of sPD-1 levels between the groups. Linear correlation analysis was performed using Spearman’s rank correlation. All tests were two-sided, and a *p*-value of < 0.05 was considered statistically significant.

### 2.5. Literature Review of Previous Studies on sPD-1 in Several Types of Cancer

Previous studies on the clinical significance of sPD-1 in several types of cancer were reviewed using the PubMed database. The search was restricted to manuscripts published in English. Search terms included “soluble PD-1”, “cancer”, and “malignancy”.

## 3. Results

### 3.1. Patients’ Clinicopathological Characteristics

Table 1 summarizes the clinicopathological characteristics of eligible patients, including changes in plasma sPD-1 concentration. To explore whether there was an association between plasma sPD-1 concentration and clinical responses in patients with cancer receiving PD-1 blockade therapy, we measured sPD-1 concentrations in plasma collected from 22 patients (12 with NSCLC, 9 with gastric cancer, and 1 with bladder cancer) at diagnosis and after two and four cycles of anti-PD-1 antibody therapy. Five patients with NSCLC and nine patients with gastric cancer were treated with nivolumab, whereas seven patients with NSCLC and one patient with bladder cancer were treated with pembrolizumab.

### 3.2. Comparison of Plasma sPD-1 Concentration at Each Treatment Point

Seven out of 22 patients had undetectable sPD-1 levels before the initiation of anti-PD-1 antibody therapy. The general information of the six healthy controls is shown in Appendix A. For the 15 patients with measurable sPD-1 levels before anti-PD-1 antibody therapy (pre-ICI), the sPD-1 values for each group are shown in Appendix A. The concentrations of sPD-1 before ICI therapy (15 cases), after two cycles of anti-PD-1 antibody therapy (14 cases), and after four cycles of anti-PD-1 antibody therapy (10 cases) were plotted with the concentrations of four healthy participants as control (Figure 1B). Since sPD-1 levels in two of the six healthy cases were below the detection limit, these data were not included in the analysis and are not represented in Figure 1B. The levels of sPD-1 significantly increased after two and four cycles of anti-PD-1 antibody therapy compared with pre-ICI levels (*p* = 0.0003 and 0.0010, respectively; Figure 1B). The administration of anti-PD-1 antibodies increased the levels of sPD-1. Moreover, we compared the sPD-1 levels in the two groups of patients who received nivolumab and pembrolizumab pre-ICI and after two and four cycles. As shown in Appendix A, sPD-1 levels after two and four cycles of nivolumab significantly increased compared with pre-ICI levels (*p* = 0.0304 and 0.0217, respectively). For pembrolizumab, sPD-1 levels after two cycles significantly increased compared with pre-ICI levels (*p* = 0.0081), but there was no significant difference between pre-ICI sPD-1 levels and those after four cycles (*p* = 0.0668).

### 3.3. Association between sPD-1 Levels and Tumor Size after Four Cycles of ICI Therapy

We were prompted to investigate whether changes in sPD-1 levels were observed in response to anti-PD-1 antibody therapy. Therefore, we calculated changes in sPD-1 concentration from baseline (pre-ICI therapy) to after two and four cycles of anti-PD-1 antibody therapy and from after two to after four cycles of anti-PD-1 antibody therapy. We also evaluated the potential association of sPD-1 levels with PFS, OS, and the percentage change in tumor size from baseline to after four cycles of anti-PD-1 antibody therapy. Table 1 shows the clinicopathological characteristics, changes in plasma sPD-1 concentrations from pre- to post-ICI therapy (two and four cycles of anti-PD-1 antibody therapy), and the relative changes in tumor sizes of 10 patients for whom sPD-1 levels after four cycles of anti-PD-1 antibody therapy were available. The changes in sPD-1 concentration from pre-ICI therapy to after two and four cycles of anti-PD-1 antibody therapy were not significantly correlated with the percentage change in tumor size (r = 0.0386 (*p* = 0.6409; Figure 2A) and 0.0022 (*p* = 0.9125; Figure 2B), respectively). Interestingly, the change in sPD-1 concentration from after two cycles to after four cycles of anti-PD-1 antibody therapy was significantly positively correlated with the percentage change in tumor size (r = 0.4881 (*p* = 0.024; Figure 2C)). In addition, we investigated the association between changes in sPD-1 levels and PFS/OS; there were no significant associations between sPD-1 concentrations and PFS/OS at pre-ICI therapy or after two and four cycles of anti-PD-1 antibody therapy (Appendix A).

## 4. Discussion

The presence of PD-L1 expression on tumor cells has been used to predict responders to anti-PD-1 antibodies in several types of cancer. However, it is not the only factor influencing the efficacy of ICIs. There are still no reliable biomarkers for predicting the therapeutic effect of ICIs. The detection of soluble forms of immune checkpoint molecules as a liquid biopsy for cancer is a novel approach to predict the efficacy of ICIs [19]. We focused on the fact that sPD-1 and sPD-L1 are present in the peripheral blood. The prognostic significance and predictive role of sPD-1 in response to ICI therapy remain unclear. This study investigated the association between plasma sPD-1 levels and the clinical efficacy of ICI therapy. We showed that the levels of sPD-1 increased in most cases after anti-PD-1 antibody therapy. Moreover, we demonstrated that the rate of change in plasma sPD-1 levels increased after two and four cycles of anti-PD-1 antibody therapy and significantly correlated with tumor progression. To the best of our knowledge, this study is the first to examine sPD-1 levels and increased tumor size for patients with cancer treated with anti-PD-1 monotherapy.

In addition to membrane-bound PD-1 on T lymphocytes, circulating sPD-1 can be detected by blood tests. sPD-1 was reported to be a monomeric protein [20]. When sPD-1 is present in the bloodstream, it correlates with the PD-1-ex3 mRNA transcript’s translational product. It is similar to membrane-bound PD-1 and is released by cleavage or as a splice variant that is no longer in the membrane-bound form [21]. However, its function and mechanism of release remain unclear. Several preclinical studies suggested that sPD-1 has a bioactive role and blocks PD-1/PD-L1 regulatory properties, which are the membrane-bound forms [22]. Currently, there are studies on the clinical significance of sPD-1 not only in cancer but also in other diseases. sPD-1 and sPD-L1 were initially reported in autoimmune diseases. Both sPD-1 and sPD-L1 are thought to be produced by immune cells upon stimulation with proinflammatory cytokines [13]. For instance, elevated levels of sPD-1 are associated with the progression of rheumatoid arthritis [23].

To date, few studies have investigated the association between the clinical efficacy of sPD-1 concentration and the prognosis of malignant tumors. We performed a literature review using the PubMed database to identify relevant articles. The results of 13 relevant studies are summarized in Table 2 [1,2,19,22,24,25,26,27,28,29,30,31,32]. Moreover, there were only three studies that investigated the clinical significance of sPD-1 in ICI-treated patients with cancer [24,25,26]. However, the prognostic role of sPD-1 was different for each type of cancer. For instance, high levels of sPD-1 in plasma or serum have been positively associated with poor clinical significance in patients with renal cell carcinoma [1,24], ovarian cancer [19], triple-negative breast cancer [27], and pancreatic ductal adenocarcinoma [29]. Conversely, sPD-1 was associated with a favorable prognosis in patients with malignant melanoma [25], NSCLC [26], and hepatocellular carcinoma [31]. Therefore, the impact of sPD-1 on prognosis and therapeutic efficacy is controversial, with no definitive opinion. In this study, we showed for the first time that an increase in the levels of plasma sPD-1 during anti-PD-1 antibody therapy was significantly correlated with tumor progression. This study suggests sPD-1 as a novel predictive biomarker for the therapeutic effect of ICIs at an early stage of treatment and provides an opportunity to elucidate the mechanism of ICI resistance.

Our results showed a significant increase in plasma sPD-1 levels after anti-PD-1 antibody therapy. Notably, these observations may address the unresolved question of the origin of sPD-1. Although the origin of sPD-1 production remains poorly understood, it has been reported that PD-1 is upregulated upon activation of T lymphocytes. Moreover, sPD-1 is essentially expressed by CD4^+^ and CD8^+^ T lymphocytes, and its expression increases following the activation of PBMCs [1,33]. At least from our results, there was no significant difference in sPD-1 levels after two and four cycles of ICI therapy (*p* = 0.7474). In other words, it is possible that the anti-PD-1 antibody does not increase the sPD-1 level but that the tumor cells and immune cells release sPD-1. Although the mechanism has not been resolved because the developmental origin of sPD-1 remains unknown, it is possible that the effect of elevating sPD-1 is more important with early anti-PD-1 antibody administration up to two cycles and is attenuated even if the administration is continued up to four cycles. We found that sPD-1 levels increased after two and four cycles of anti-PD-1 antibody therapy compared with pre-ICI therapy levels. We attribute these results to the activation of PBMCs, including T lymphocytes, by anti-PD-1 antibody therapy. Moreover, it was reported that an induced or augmented increase in PD-1 expression might suppress the immune response of T lymphocytes, resulting in increased sPD-1 expression [28]. Additionally, previous studies have shown that sPD-1 level is elevated in patients with cancer compared with that in healthy subjects. Therefore, sPD-1 may be used as a diagnostic biomarker in future studies.

The function of sPD-1 is debated. The membrane-bound PD-1 receptor on T cells is reportedly left inactivated, and the inhibitory signal is reduced when sPD-1 blocks PD-L1 on tumor cells [22]. Elhag et al. [34] reported an elevated sPD-1 concentration related to prolonged survival in a tumor-bearing murine model via reduced immunosuppression. In particular, sPD-1 increases cytotoxicity and reduces tumor-infiltrating lymphocyte suppression of T lymphocytes [34]. Moreover, there are reports showing that sPD-1 can inhibit all three PD-L1/PD-1 interactions: PD-L1/CD80, PD-L1/PD-1, and PD-L2/PD-1 [35]. In other words, sPD-1 leads to the inhibition of PD-L1 by binding to the ligand so that PD-L1 cannot bind to PD-1 on T cells. An in vivo study confirmed that sPD-1 competed with PD-1 and reduced interleukin-10 expression [36]. Higher levels of plasma sPD-1 may be related to an active anticancer immune response, which is suppressed in the presence of tumors. Therefore, we expected that sPD-1 levels would increase in response to an excellent clinical response to anti-PD-1 antibody therapy. However, our results show that elevated levels of sPD-1 during anti-PD-1 antibody therapy correlated with tumor progression. sPD-1 may be involved in resistance to anti-PD-1 antibody therapy, suggesting that changes in sPD-1 concentrations could identify primary ICI non-responders early in treatment.

We observed that sPD-1 levels significantly increased in association with tumor progression. Potential mechanisms by which elevated levels of sPD-1 may contribute to tumor growth are described as follows. Anti-PD-1 antibodies, such as nivolumab and pembrolizumab, have an affinity toward sPD-1, and sPD-1 could be bound to the drugs in an sPD-1/anti-PD-1 antibody complex. We speculated that the complex inhibits anti-PD-1 antibody binding to PD-1 and correlates with resistance to PD-1 blockade therapy. In other words, sPD-1 may act as a decoy, resulting in reduced circulatory clearance of anti-PD-1 antibodies [25].

There are several limitations to this study. First, our study was retrospective in nature and had a small sample size. To evaluate the relationship between ICI and sPD-1 and the clinical significance of sPD-1 levels in patients receiving ICI therapy, this study focused on monotherapy with anti-PD-1 antibodies. The reason for the small sample size was that lung cancer, gastric cancer, and bladder cancer are now mainly treated with combination therapy consisting of immune checkpoint and cytotoxic chemotherapy agents rather than monotherapy. Therefore, it is difficult to collect data from patients who are treated with anti-PD-1 monotherapy. Second, although it is noteworthy that all studies were conducted in patient populations receiving anti-PD-1 antibody therapy, multiple types of cancer were included. Therefore, no significant associations between sPD-1 levels and PFS/OS were observed. We believe that the change in tumor size observed during anti-PD-1 therapy was an appropriate indicator of the treatment efficacy of ICIs. Further prospective studies with larger sample sizes and longer follow-up periods should be conducted to verify the novel role of sPD-1 in patients with cancer. In addition, we believe that by increasing the number of samples, it will be possible to conduct analyses limited to each type of cancer. Third, this study showed that the sPD-1 level before anti-PD-1 antibody administration was low, and it would be difficult for the sPD-1 level to be a biomarker for predicting treatment effect before starting anti-PD-1 antibody therapy. Thus, we decided to focus on the rate of change in sPD-1 levels based on the result that sPD-1 increased after ICI administration. We thought that analyzing the relationship between the rate of change after anti-PD-1 antibody administration and tumor size would lead to a faster prediction of the treatment effect of ICIs.

## 5. Conclusions

A significant correlation between increased levels of sPD-1 and tumor progression was observed during anti-PD-1 antibody therapy in patients with NSCLC, gastric cancer, and bladder cancer. Importantly, our samples were all collected from patients administered anti-PD-1 antibody therapy (nivolumab or pembrolizumab). Our data suggest that sPD-1 is involved in resistance to anti-PD-1 antibody therapy, suggesting that changes in sPD-1 concentrations during anti-PD-1 antibody therapy could identify primary ICI non-responders early in treatment. We also showed a significant increase in plasma sPD-1 levels after anti-PD-1 antibody therapy. These results suggest that sPD-1 levels correlate with resistance to PD-1 blockade therapy. However, the function and origin of sPD-1 have not been thoroughly investigated. Furthermore, analyzing each specific type of cancer and evaluating a larger sample size may provide new insights into the clinical significance of sPD-1 and clarify the predictive role of sPD-1 in patients treated with ICIs.

## Figures and Tables

**Figure 1 biomedicines-09-01929-f001:**
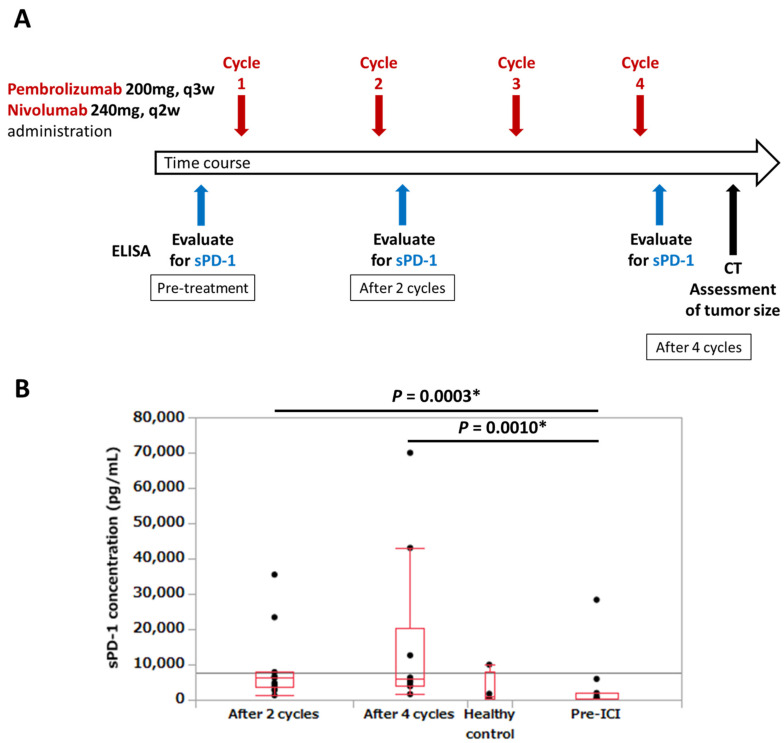
Plasma sPD-1 concentrations at each treatment point. (**A**) Schema for schedule of ICI administration and timing of sPD-1 measurement. We retrospectively analyzed data from 22 patients (12 with first-line or previously treated NSCLC, 9 with gastric cancer, and 1 with bladder cancer) who received anti-PD-1 antibody therapy (nivolumab (240 mg) intravenously every 2 weeks or pembrolizumab (200 mg) intravenously every 3 weeks). Plasma levels of sPD-1 were evaluated pretreatment and after two and four cycles of anti-PD-1 antibody therapy. A change in tumor size was defined as the percentage change in tumor size from baseline to after four cycles of anti-PD-1 antibody therapy. (**B**) Blood samples were collected before and after treatment, and the plasma levels of sPD-1 were measured by en-zyme-linked immunosorbent assay (ELISA). We measured sPD-1 concentration for healthy control subjects (N = 4), and patients administered anti-PD-1 antibodies. For patients, sPD-1 was measured before treatment (N = 15), after 2 cycles (N = 14), and at the point after 4 cycles (N = 10) of ICI administration. We plotted the sPD-1 concentration at each time point and compared each. The levels of sPD-1 were significantly increased after 2 and 4 cycles, compared to pretreat-ment levels (*p* = 0.0003; *p* = 0.0010, respectively). * Statistically significant.

**Figure 2 biomedicines-09-01929-f002:**
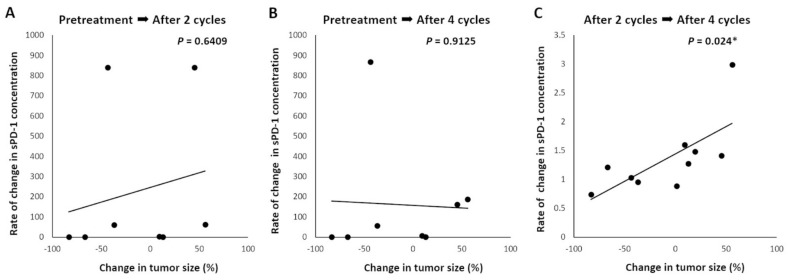
Linear correlation between change in plasma level of sPD-1 and change in tumor size. Relative changes in sPD-1 were calculated as the concentration from the baseline (pre-ICI treatment) to after two and four cycles and two to four ICI treatment cycles. Then, we evaluated its potential association with the change in tumor size from the baseline to after 4 cycles. The change in sPD-1 concentration from pre-ICI to after both two and four cycles was not significantly correlated with the percentage change in tumor size (r = 0.0386, *p* = 0.6409, (**A**); r = 0.0022, *p* = 0.9125, (**B**), respectively). The changes in sPD-1 concentration from after 2 cycles to after four cycles were positively correlated with the percentage change in tumor size with statistical significance (r = 0.4881, *p* = 0.024; (**C**)). * Statistically significant.

**Table 1 biomedicines-09-01929-t001:** Clinicopathological features, plasma soluble programmed death protein 1 (sPD-1) concentration, relative change in tumor size, and PFS/OS for all study patients.

Case	Sex	Age, Years	Cancer Type	Stage	ICI	Tumor mPD-L1 Expression (IHC,%)	sPD-1 Concentration (pg/mL)						Relative Change in Tumor Size (%)	PFS (Months)	OS (Months)
							Pre-ICI	Post-ICI		Relative Change					
								After 2 Cycles	After 4 Cycles	After 2 Cycles/Pre-ICI	After 4 Cycles/Pre-ICI	After 4 Cycles/After 2 Cycles			
1	M	78	NSCLC	IV	Nivolumab	20~30	110.20	6642.70	6338.05	60.28	57.51	0.95	-36.9	40.4	45.0
2	M	61	NSCLC	III^R^	Nivolumab	N/A	5940.23	6659.75	4947.05	1.12	0.83	0.74	-83.3	20.3	28.6
3	F	70	NSCLC	IV	Nivolumab	N/A	N.D.	2757.00	4093.38	N/A	N/A	1.48	19.4	2.3	5.6
4	M	67	NSCLC	III^R^	Nivolumab	70~80	28,352.41	35,479.60	43,041.72	1.25	1.52	1.21	-66.9	13.5	27.2
5	M	63	GC	III^R^	Nivolumab	N/A	33.30	3796.888	5369.81	839.28	161.27	1.41	45.3	3.4	4.2
6	M	74	GC	IV	Nivolumab	N/A	421.07	N/A	N/A	N/A	N/A	N/A	N/A	2.4	4.8
7	M	68	GC	IV	Nivolumab	N/A	1959.76	7870.94	12,614.14	4.02	6.44	1.60	9.0	3.9	8.3
8	M	67	NSCLC	III^R^	Nivolumab	50~60	1035.29	1276.05	1625.91	1.23	1.57	1.27	12.7	1.3	8.2
9	F	68	GC	IV	Nivolumab	N/A	N.D.	4355.20	3871.38	N/A	N/A	0.89	1.1	1.3	2.3
10	M	66	GC	III^R^	Nivolumab	N/A	N.D.	2945.92	N/A	N/A	N/A	N/A	N/A	1.2	7.4
11	M	60	GC	III^R^	Nivolumab	N/A	N.D.	N/A	N/A	N/A	N/A	N/A	N/A	0.5	1.5
12	F	49	GC	III^R^	Nivolumab	N/A	386.35	N/A	N/A	N/A	N/A	N/A	N/A	0.5	2.5
13	F	75	GC	IV	Nivolumab	N/A	N.D.	N/A	N/A	N/A	N/A	N/A	N/A	3.6	9.4
14	F	57	GC	IV	Nivolumab	N/A	245.33	N/A	N/A	N/A	N/A	N/A	N/A	1.2	1.5
15	M	72	NSCLC	IV	Pembrolizumab	70~80	373.86	23,400.47	69,979.36	62.59	187.18	2.99	56.0	2.4	7.5
16	M	71	NSCLC	IV	Pembrolizumab	60~70	378.87	N/A	N/A	N/A	N/A	N/A	N/A	20.4	34.3
17	M	59	NSCLC	IV	Pembrolizumab	60~70	40.85	N/A	N/A	N/A	N/A	N/A	N/A	1.0	1.5
18	M	64	NSCLC	IV	Pembrolizumab	60~70	7.42	6225.54	6441.62	839.28	868.41	1.03	-43.6	7.0	22.0
19	M	70	NSCLC	IV	Pembrolizumab	70~80	426.00	6414.20	N/A	N/A	N/A	N/A	N/A	1.6	1.7
20	M	71	NSCLC	IV	Pembrolizumab	>90	N.D.	4840.64	N/A	N/A	N/A	N/A	N/A	2.4	4.6
21	F	70	BLDC	IV	Pembrolizumab	N/A	1967.59	8023.49	N/A	N/A	N/A	N/A	N/A	7.4	20.3
22	M	68	NSCLC	IV	Pembrolizumab	10~20	N.D.	N/A	N/A	N/A	N/A	N/A	N/A	6.1	26.7

ICI; immune checkpoint inhibitor., PFS; progression-free survival, OS; overall survival, F; female, M; male, NSCLC; non-small-cell lung cancer, GC; gastric cancer, BLDC; bladder cancer, R; recurrence, N/A; not applicable or available, N.D.; not detected. N.D. indicates a result below the method detection limit.

**Table 2 biomedicines-09-01929-t002:** Previous studies of clinical significance of sPD-1 in several types of cancers.

	Author, Year of Publication	Tumor Types	Patients Number	Serum/Plasma	ICI	Major Findings Related to sPD-1
1	Montemagno C, et al., 2020	RCC	50 (Sunitinib) 37 (Bevasizumab)	Plasma	—	High levels of sPD-1 were independent prognostic factors of PFS in the sunitinib group. (The levels of sPD-1 were not correlated to PFS under bevacizumab.)
2	Incorvaia L, et al., 2020	RCC	9 (long-responder)	Plasma	Nivolumab	At baseline, high sPD-1 levels were observed. Conversely, after 4 weeks from starting nivolumab, sPD-1 levels were strongly reduced only in patients with PR/CR/SD to nivolumab >18 months.
3	Pawłowska A, et al., 2020	OC	50	Plasma	—	The higher level of CD4^+^PD-1^+^ T cells in the circulation and the higher sPD-1 level in plasma predict poor survival of OC patients.
4	Babačić H, et al., 2020	MM	24	Plasma	Nivolumab Ipilimumab Ipi + Nivo	Circulating sPD-1 had the highest increase during anti-PD-1 treatment and in anti-PD-1 responders.
5	He J, et al., 2020	NSCLC	88	Plasma	—	The plasma concentrations of sPD-1 were higher than those in the healthy control group. Higher sPD-L1/sPD-1 ratio indicates a relatively better prognosis. (High levels of sPD-L1 indicates better prognosis, but the levels of sPD-1 were not correlated to survival time.)
6	Tiako Meyo M, et al., 2020	NSCLC	87	Serum	Nivolumab	After two cycles of nivolumab, an increased or stable sPD-1 level independently correlated with longer PFS and OS).
7	Li Y, et al., 2019	TNBC	59	Serum	—	Compared to healthy women, the serum concentration of sPD-1 was significantly elevated in TNBC patients. Patients who experienced complete or partial remission after NAC had significantly decreased serum levels of sPD-1 compared to patients with a poor response to NAC.
8	Dillman RO, et al., 2019	MM	39	Serum	—	Baseline sPD-1 (cut-off value is 1,200 pg/mL) was not a prognostic marker for survival for melanoma patients.
9	Bian B, et al., 2019	PDAC	32	Plasma	—	The soluble forms of PD-1 and PD-L1 share a strong correlation. Patients with high level of sPD-1 (>8.6 ng/ml) have a shorter overall survival than for patients with low level of sPD-1.
10	Tominaga T, et al., 2019	CRC	117	Serum	—	The concentrations of sPD-1 both pre- and post-CRT were not associated with DFS.
11	Chang B, et al., 2019	HCC	120	Serum	—	The level of sPD-L1 positively correlated with the level of sPD-1. The high level of sPD-1 correlated with a favorable OS, as well as a trend toward prolonged DFS.
12	Kruger S, et al., 2017	PDAC	41	Serum	—	The close correlation was observed between levels of sPD-1 and sPD-L1. To compare OS in patients with high vs. low sPD-1 and sPD-L1 serum levels, both sPD-1 and sPD-L1 levels did not indicate an adverse outcome.
13	Sorensen SF, et al., 2016	NSCLC	38	Serum	—	The serum concentration of sPD-1 was found to be significantly higher at disease progression as compared to pre-treatment. An increase in sPD-1 during treatment was associated with prolonged progression-free survival and overall survival.

RCC; renal cell carcinoma, OC; ovarian cancer, MM; malignant melanoma, NSCLC; non-small-cell lung cancer, TNBC; triple negative breast cancer, PDAC; pancreatic ductal adenocarcinoma, CRC; colorectal cancer, HCC; hepatocellular carcinoma.

## Data Availability

The data presented in this study are available on request from the corresponding author.

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
