# Peer review of "Increased Plasma Soluble PD-1 Concentration Correlates with Disease Progression in Patients with Cancer Treated with Anti-PD-1 Antibodies"

_biomedicines, 2021, doi:10.3390/biomedicines9121929_

Round 1

Reviewer 1 Report

  • Introduction section should be revised by the authors and modified to make it more readable and concise for the readers. For example, in my opinion, second paragraph should be the first and only include essential information presented in the currently first paragraph.
  • I do not see clear description of the cancer-related mechanism in which immune response is avoided by these cells through PD-1/PD-L1-axis. Moreover, considering the fact that the manuscript focus on the lung NSCLC and gastric cancer, information of these mechanisms should be described also directly in context of these cancers.
  • The authors might want to provide more exact information on the patients selection for anti-PD-1/anti-PD-L1 therapy.
  • Regarding patients characteristic in ‘Materials and methods’ section, information on how subjects were qualified for anti-PD1 therapy is missing. Those should include for example histopathological results demonstrating whether cancers demonstrated higher expression of PD-L1 on their surface.
  • The authors should reconsider increasing patients number included in the study as right now it is hard to even consider demonstrated results as highly significant in context of whole population. Moreover, I presume that combination of NSCLC, gastric and bladder cancer patients was associated predominantly with low number of subjects in each group separately. If the authors would manage to increase these groups then I would recommend to show changes in plasma sPD-1 for each study group separately.
  • As the analyses are performed on the plasma, it is hard to understand why only four healthy controls were collected by the authors. In addition, these patients should also be characterized in context of sex, age, immunological status etc.
  • As currently there is no problem with figures numbers within the manuscript, I think that visual presentation of the therapeutic protocol could be a part of Figure 1.
  • The description of the statistical analyses should be improved. There is no information on the data distribution, exact tests used. In addition, considering small number of subjects it is hard to believe that these data could have Gaussian distribution, therefore, obtained results should be analyzed using non-parametric tests and presented as medians and interquartile ranges instead of means and standard deviations.
  • I do not see the point of literature review in that original paper. These information should be used for discussion of the results but there is no point in presenting them as part of results section.
  • The description of the histopathological part should be extended. Moreover, I did not see results from these diagnostic procedures within the ‘Results’ section.
  • Regarding the ‘Results’ section, as indicated above the numbers of subjects should be increased, also in the context of healthy control group. At the current moment, because of small healthy control subjects, it is not even possible to perform reliable statistical comparison between that group and cancer patients groups.
  • If the authors will manage to improve group size, it would be beneficial to analyze association between sPD-1 and tumor size in each type of cancer separately. Additionally, why the authors use rate of change in sPD-1 instead of just analyzing correlations at each therapy stage separately.
  • I presume that the authors might have access to the outcome of the anti-PD-1 treatment, thus, analyzes of survival in context of soluble PD-1 should be performed. Moreover, as it was suggested that sPD-1 might be helpful in identifying patients not responding to the anti-PD-1 therapy, the authors should also prepare results that could justify that point of view.
  • Considering critical limitations associated with sample size and data missing, it is hard to evaluate the discussion of currently presented data. If authors will manage to perform all the necessary corrections within the manuscript then the discussion section will be completely revised.
  • The whole manuscript should be verified by native speaker or advanced level English speaker experienced in biomedical texts as there are numerous structural and spelling errors within the text.

Reviewer 2 Report

Tha manuscirpt by Ohkuma and colleagues aims to analyse the role of sPD-1 in cancer patients udergoing anti-PD-1 therapy. Considering the rising importance of soluble form of immune checkpoint inhibitors the topic of the paper is of high interst. However, few point should be addressed before considering it for publication.

1) Are there any differences in sPD-1 levels between patients treated with ivolumab or pembrolizumab?

2) According to Fig1 no differences in sPD-1 level could be detected after2 and 4 cycles, as if a plateau has been reached. Authors should discuss this point

Reviewer 3 Report

In this manuscript, the authors have investigated the clinical implications of soluble anti-programmed death-1 (sPD-1) plasma levels in cancer patients receiving immune checkpoint inhibitors. The authors found that after 2–4 cycles of anti-PD-1 antibody therapy, there was a higher rate of change in plasma sPD-1 levels, which was substantially linked with tumor progression. Based on the data, the authors have concluded that the Changes in sPD-1 may be able to detect primary ICI non-responders early in treatment, suggesting that it is implicated in resistance to anti-PD-1 antibody therapy. This is an interesting study and has clinical relevance. However, the following points should be addressed.

  1. Immunohistochemical images of PD-1 should be provided.
  2. The association between IHC staining and clinicopathological feature is needed.
  3. The novelty of the study should be clearly emphasized.

Round 2

Reviewer 1 Report

The authors managed to respond to all of my comments.

Reviewer 2 Report

Authors fully responded to reviewer comments  and the manuscript has beensufficiently improved to warrant publication.

Reviewer 3 Report

The authors have addressed all the comments and the manuscript can be accepted for publication.